# Margin Free Resection Achieves Excellent Long Term Outcomes in Parathyroid Cancer

**DOI:** 10.3390/cancers15010199

**Published:** 2022-12-29

**Authors:** Klaus-Martin Schulte, Nadia Talat, Gabriele Galatá

**Affiliations:** 1Academic Department of Surgery, School of Medicine and Psychology, College of Health and Medicine, Australian National University, Canberra, ACT 2600, Australia; 2Department of Endocrine Surgery, King’s College Hospital, King’s Health Partners, London SE5 9RS, UK

**Keywords:** parathyroid cancer, parathyroid carcinoma, en bloc resection, margin status, survival, long-term outcomes, lymphadenectomy, compartment, recurrent laryngeal nerve

## Abstract

**Simple Summary:**

Long-term outcomes of parathyroid cancer are unsatisfactory with common locoregional recurrence and significant mortality. Our case series provides evidence that an oncological surgical approach lastingly benefits patients, affording a 10-year disease-specific survival rate of 100%. Core principles are pre-operative recognition of potential malignancy, en-bloc resection ensuring cancer free resection margins (R0) and clearance of the central lymph node compartment, followed by initially dense follow-up for timely detection and aggressive treatment of recurrence.

**Abstract:**

Long-term outcomes of parathyroid cancer remain poorly documented and unsatisfactory. This cohort includes 25 consecutive parathyroid cancer patients with median follow-up of 10.7 years (range 4.1–26.5 years). Pre-operative work-up in the center identified a suspicion of parathyroid cancer in 17 patients. En bloc resection, including the recurrent laryngeal nerve in 4/17 (23.5%), achieved cancer-free resection margins (R0) in 82.4% and lasting loco-regional disease control in 94.1%. Including patients referred after initial surgery elsewhere, R0 resection was achieved in merely 17/25 (68.0%) of patients. Cancer-positive margins (R1) in 8 patients led to local recurrence in 50%. On multivariate analysis, only margin status prevailed as independent predictor of recurrence free survival (χ^2^ 19.5, *p* < 0.001). Local excision alone carried a 3.5-fold higher risk of positive margins than en bloc resection (CI_95_: 1.1–11.3; *p* = 0.03), and a 6.4-fold higher risk of locoregional recurrence (CI_95_: 0.8–52.1; *p* = 0.08). R1-status was associated with an 18.0-fold higher risk of recurrence and redo surgery (CI_95_: 1.1–299.0; *p* = 0.04), and a 22.0-fold higher probability of radiation (CI_95_: 1.4–355.5; *p* = 0.03). In patients at risk, adjuvant radiation reduced the actuarial risk of locoregional recurrence (*p* = 0.05). When pre-operative scrutiny resulted in upfront oncological surgery achieving cancer free margins, it afforded 100% recurrence free survival at 5- and 10-year follow-up, whilst failure to achieve clear margins caused significant burden by outpatient admissions (176 vs. 4 days; χ^2^ 980, *p* < 0.001) and exposure to causes for concern (1369 vs. 0 days; χ^2^ 11.3, *p* = 0.003). Although limited by cohort size, our study emphasizes the paradigm of getting it right the first time as key to improve survivorship in a cancer with excellent long-term prognosis.

## 1. Introduction

Parathyroid cancer is a rare endocrine neoplasm with an incidence rate of about 1 case per million per year [1]. It often presents with severe hypercalcemia, excessive PTH elevation and systemic complications [2,3,4]. Historically, the prognosis of parathyroid cancer has been guarded. Failure of locoregional disease control results in recurrence of hypercalcemia, only tardily and uncommonly followed by systemic spread [5]. Death due to parathyroid cancer is mainly driven by recalcitrant hypercalcemia and its systemic repercussions rather than structural disease [6].

In the past, the academic interest in parathyroid cancer has often focused on survival. The 10-year survival rate of parathyroid cancer has historically been poor, with less than 75% of patients surviving at 10 years [5,7,8,9]. Despite the relatively young mean age at the initial presentation of 49–62 years [5,9,10,11], parathyroid cancer belongs to those malignancies where the cause of death is more commonly due to cardiac or other causes than cancer itself [12]. Whilst data from the SEER register reveal a 10-year overall survival (OS) of only 67.1%, disease-specific survival is estimated to be 92.1% [13]. In the age group at risk, such a net difference of 25% excess death may relate to the increased cardiovascular risk related to chronic or recurrent hyperparathyroidism and hypercalcemia [14], particularly when they are prolonged or excessive [15]. The multiple facets of chronic hypercalcemia and its long-term complications have recently been revisited [16]. 

From a patient perspective, adverse experience arises from failure to cure the cancer during the initial surgery, necessitating repeat surgical interventions and adjuvant therapies such as radiation, both bringing along a significant treatment burden. However, series from leading surgical centers revealed that 49% of patients suffered recurrence, with 60% suffering lasting treatment associated complications [9]. Failing to eradicate the root cause, hypercalcemia is not easily steered [17,18].

Against that backdrop, treatment goals in parathyroid cancer include not only the prevention of cancer related mortality, but the lasting restoration of calcium homeostasis, in line with the guideline surgical approach to primary hyperparathyroidism [19]. Upfront cure, rather than attempts at rescue, preempts all of the harrowing consequences of parathyroid malignancy. We here present data to support the argument that not just cancer-specific survival, but improved survivorship through avoidance of paraneoplastic hypercalcemia and treatment-related complications endorse the vigorous attempt to achieve oncological control in the initial stage of disease.

There is little debate that parathyroid cancer should initially be treated by surgery [20], with radiation therapy and systemic chemotherapy presenting adjuvant but unproven options [20,21]. However, a fatalistic approach persists regarding our ability to recognize the diagnosis and deliver appropriate surgery from the start: failing to discern parathyroid carcinoma from adenoma preoperatively, the diagnosis of carcinoma is often made only after parathyroidectomy [21]. Cure is achieved in only 50% of patients, following what is yet perceived as appropriate initial surgical management [22].

We had previously reported a cohort of 19 patients with parathyroid cancer: those treated by oncological resection rather than just local excision exhibited significantly improved disease-free survival, though at limited follow-up [23]. The current publication includes six additional patients and confirms the important observation that upfront oncological surgery improves survivorship by lasting loco-regional disease control. Long-term overall and recurrence-free survival beyond 10 years was observed in all patients in whom cancer-free margins in terms of an R0 resection were achieved during either initial surgery or upon cancer recurrence.

## 2. Materials and Methods

Consistent with our prior reported approach [23], we retrospectively analysed data from now 25 consecutive patients with a histological diagnosis of parathyroid cancer treated in our tertiary endocrine surgery centre between 1996 and 2022. One patient was operated at the centre in 1996 by another surgeon (Appendix A). Four patients underwent surgery (local excision) elsewhere and were referred in for the unexpected cancer diagnosis. One of these had persistent disease at the time of referral as the prior targeted parathyroidectomy had failed to identify a target (Patient 18). Two of these patients developed loco-regional recurrence and underwent redo surgery. Twenty parathyroid cancer patients were primarily operated upon by the lead author (KMS) between 2005–2018 (Appendix A). Cancer had been suspected in all but three patients (85%), and these 17 patients underwent oncological resection with en bloc removal of the parathyroid lesion with surrounding fat tissue, ipsilateral thyroid lobe and level VI lymph nodes (Appendix A). In the earlier part of the series the ipsilateral lateral lymph nodes were also included. A total of 271 lymph nodes were excised in 25 patients of which 161/271 were from the central compartment level VI and 110/271 were from levels II–V (Appendix A). The recurrent laryngeal nerve was resected in 4 patients (20%) because it was visibly involved by the cancer or could not be spared without risk to disrupt the integrity of specimen margins. The 3 patients who underwent local excision only had calcium levels <3.0 mM and lesion sized less than 30 mm.

Surgical intervention was performed as defined previously [3,24,25]:-Local excision (LE) only, i.e. pericapsular excision of the parathyroid lesion such as that usually employed for benign lesions.-En bloc excision; this includes en bloc and oncological resection. En bloc describes excision of the parathyroid with circumferential soft tissue as minimal criterion; in this process the tumour capsule must not be laid open at any point; oncological resection additionally includes ipsilateral thyroid lobectomy, centro-cervical lymphadenectomy of level VI lymph nodes and/or further locoregional excision.Histological diagnosis was defined by WHO criteria [26].Resection Margins were defined as per WHO criteria [26].-R0: no cancer cells at the margins.-R1: cancer cells at the edge of the histological specimen or resection within less than 1 mm of the edge.

Patients were also classified as low-risk or high risk according to the validated parathyroid cancer classification scheme described previously [25]:-Low risk: Capsular invasion combined with invasion of surrounding tissue only.-High risk: Vascular invasion and/or lymph node metastases and/or invasion of vital organs and/or distant metastases.

Consistent with our prior reported approach [23], follow-up of patients was achieved through a dedicated parathyroid cancer clinic with clinical review, and assessment of corrected calcium and PTH every 3 months. Imaging was obtained in presence of biochemical signs of cancer recurrence.

Values for mean, median, Chi square test results, and Kaplan–Meier statistics were calculated using SPSS software version 18.0.

### 2.1. Literature Review

The NIH database PubMed (http://www.ncbi.nlm.nih.gov/sites/entrez accessed on 8 August 2022) was searched using the search terms ‘parathyroid’ and ‘cancer’. Articles published between 2010–2022 were included. The search identified 4360 articles. For historic data we used data from our previous study on parathyroid cancer [5].

### 2.2. Eligibility Criteria

Articles published in English, French, German and Spanish were included. After review of the title, abstract and keywords, 30 articles were selected (Table 8, Appendix A). We only included Single Centre case series and Registry data for analysis. Individual case reports were excluded. Articles with insufficient follow up data and duplicate large Registry data were also excluded. The PRISMA flow diagram (Appendix A) provides details. 

## 3. Results

We identified 24 consecutive patients with parathyroid cancer from a prospective in-house register of endocrine malignancy. One additional patient had undergone treatment by our predecessors and was identified during a clinic visit. The total cohort comprised 25 patients (Appendix A).

All specimens were reviewed by an experienced endocrine pathologist (SDC) in order to confirm a diagnosis of parathyroid cancer and assess margin status. He also classified cancers according to the Schulte classification [3,5,25] as low risk 13/25 (52%) or high risk 12/25 (48%).

### 3.1. Surgical Procedures, Margin Status and Nodal Status

Twenty patients underwent initial surgery by the lead author. 

Pre-operative work-up correctly identified 17 of these (85.0%) as potential parathyroid cancer. These 17 (100%) underwent oncological resection with en bloc removal of the parathyroid lesion with surrounding fat tissue, ipsilateral thyroid lobe and level VI lymph nodes (Appendix A). The recurrent laryngeal nerve was resected in 4 patients (23.5%) because it was tangibly involved by the cancer or could not be spared without risk to disrupt the integrity of specimen margins. R0 margins were achieved in 3/4 (75%) of these patients. A total of 271 lymph nodes were excised in these 17 patients (Appendix A) with a total of 2/271 (0.7%) resected lymph nodes positive across the cohort. 2/161 (1.2%) lymph nodes of the central compartment level VI were positive, and 0/110 from the lateral compartments II–V [27]. Only one patient in our cohort (5.9%) exhibited a positive nodal status (No 2).

One patient developed combined loco-regional and systemic recurrence (No 2). During his initial surgery, we excised a broad disc of full-thickness hypopharynx and sacrificed the invaded RLN, yet still achieved only R1 status. One further patient (No 13) exhibited an R1 resection margin (within 1 mm of edge) despite en bloc resection, yet PTH and calcium levels normalized, and he received 60 Gy of EBRT. He remained recurrence free at 10.0 years of follow-up. Pre-operative work-up and intra-operative assessment failed to identify a cancer suspicion in three patients (No 12, 22, 24). They underwent local excision alone, 1/3 (33.3%) was margin-positive R1, here defined by cancer within 1 mm of the resection edge, and all remained free of recurrence at follow-up lasting 5.1–10.1 years.

Five patients underwent initial surgery by other surgeons: 5/5 (100%) underwent local excision, resulting in an R1 margin status in 4/5 (80.0%).

### 3.2. Outcomes of Follow-Up

Follow-up information included death, cause of death, and recurrence status. No patient was lost to follow-up. Three-year follow-up is available for all patients (*n* = 25; 100%). Four patients exhibited persistent (*n* = 1) or recurrent (*n* = 3) disease. One patient (No 21) died from an unrelated cause after 14 months of FU, and three further patients died from other causes at 63, 83, and 123 months of FU. Five-year follow-up was mature in 24/25 patients (96.0%) with one patient (No 25) followed up for 50 months only. 10-year follow-up information (death, cause of death, recurrence) was mature for 19/25 patients (76.0%).

#### 3.2.1. Overall Survival (OS)

For the mature 5-year follow-up cohort (*n* = 24) with a mean age of 58.0 ± 13.6 years at diagnosis, 5-year OS was 23/24 (95.8%). For the mature 10-year cohort (*n* = 19) with a mean age of 58.0 ± 11.8 years at diagnosis, 10-year OS was 16/19 (84.2%). Kaplan–Meier-analysis did not reveal a difference in OS between patients with R0 and R1 status: χ^2^ = 0.0037, with a 95% confidence interval of [−∞–1.0] and *p* = 0.95.

#### 3.2.2. Recurrence-Free Survival (RFS), Distant Metastasis, and Disease-Specific Survival (DSS)

The 5-year RFS was 19/23 (82.6%). The 10-year RFS was 15/19 (79.0%). The 5-year and 10-year DSS was 23/23 (100%) and 19/19 (100%), respectively. One patient with local recurrence at 1.1 years post initial surgery also presented with distant metastasis (multiple lung metastases) at 4.8 years, and died due to disease at 10.9 years (No 2). The number of events is too low to calculate DSS (a priori power 0.09).

### 3.3. Recurrence and Treatment Outcomes after Recurrence

Four patients were not cured following initial surgery, whereof one with persistent and three with recurrent disease. All cases of persistence/recurrence were observed within 3 years of the initial surgery (Figure 1). All patients underwent redo surgery with curative intent, lastingly normalising PTH and calcium in 3 out of 4 patients (75%) with a follow-up of 5.0, 8.4, and 8.6 years after revision surgery. Following multidisciplinary team discussion, three patients also received 55–60 Gy external beam radiation therapy EBRT applied in 20–30 fractions as consolidation therapy. One of these 17 patients (No 2) presented with a locally highly aggressive cancer with infiltration of hypopharynx/esophagus and nodal metastatic disease in central and lateral compartment (2/31 nodes). Extensive surgery with primary oropharyngeal/esophageal repair still resulted in R1 resection with microscopic cancer cells at the resection edge, recurrent nerve palsy due to nerve resection, and local infection with drainage needs due to an assumed anastomotic leak. He received 55 Gy of EBRT, yet went on to develop lung metastases at 2.0 years following the revision surgery and died 10.9 years following the initial diagnosis.

### 3.4. Surgery as Driver of Margin Status and Recurrence 

Baseline characteristics did not differ between the subcohorts in whom R0 or R1 resection was achieved. In particular, patients ending up with an R1 status did not exhibit larger or biochemically more active lesions, nor more aggressive histological features as classified by Schulte’s high risk/low risk classification [25] (Table 1).

Regression analysis identified margin status, but not any other clinical factor as predictor of recurrence free survival (Table 2).

The choice of the initial surgical procedure was a strong risk predictor for pathological margin outcomes (Table 3).

Pathological margin outcome was a strong risk predictor for loco-regional recurrence (Table 4).

### 3.5. Locoregional Outcomes following Adjuvant Radiotherapy

Adjuvant radiotherapy was provided to 5/8 (62.5%) patients with R1 resection, whereof in two cases after initial surgery and in 3 only following recurrence (Appendix A). Four locoregional recurrences were observed across the 8 patients initially at risk (R1 status), adding up to 12 scenarios at risk of recurrence (R1 status or status post redo surgery). Following EBRT, all of remained free of recurrence at long term follow-up (10.2, 8.9, 8.6, 8.3, and 5.0 years after EBRT) (Appendix A). Adjuvant radiotherapy tendentially reduced the risk of recurrence in scenarios at risk (Table 5).

In patients at increased risk of locoregional recurrence, such risk being identified as either a positive margin status after initial surgery or status post redo surgery, adjuvant external beam radiotherapy EBRT achieved a significant improvement of locoregional recurrence-free survival (Figure 2).

### 3.6. Survivorship of Patients with Parathyroid Cancer

The majority of patients remained free of recurrence, required no further cancer-related hospital admissions, nor any specific medication beyond vitamin D supplementation. However, the therapy burden strongly differed between patients in whom cancer-free margins were achieved and those in whom R1 resections were followed by a need for further surgery, adjuvant radiotherapy, and eventual recurrence (Table 6).

### 3.7. Benchmarking Outcomes

#### 3.7.1. Survival Outcomes

In our cohort, disease-specific survival was 100% at 10 years. A single patient died from disease during follow-up beyond 10 years.

Table 7 provides an overview of relevant survival data in published case series by peers and national databases. All survival parameters in this cohort compare favorably to the median of prior published outcomes in small case series (Appendix A) and in National cohorts from large databases (Appendix A).

#### 3.7.2. Outcomes following Adjuvant Radiotherapy

As Table 5 and Figure 2 suggest a benefit of EBRT in high-risk scenarios, we sought to benchmark its use by review of EBRT related outcomes in PC in the published literature. 123 patients from 10 small case series were analysed in whom adjuvant radiotherapy was given after initial parathyroid cancer surgery (Table 8 and Appendix A). Of 17/52 patients who underwent radiotherapy after initial surgery, 32.7% suffered locoregional recurrence, whereas local recurrence occurred in 45.1% in the non-radiotherapy group, with a relative risk of 0.7 (95% CI 0.5–1.2; *p* < 0.18), also see Table 8.

## 4. Discussion

We report outcomes in a single center case series of parathyroid cancer. In rare conditions like parathyroid cancer, large register studies provide a reasonable framework to define population wide outcome expectations [7,11,13,51,52,53,58,59,60,61]. In absence of prospective evidence from randomized controlled studies due to the extraordinary rarity of the condition even in centers of excellence [62], the granular detail of case series is crucial to understand the impact of specific treatment approaches. Whilst such evidence may be insufficient to inform formal guidelines [20], it will still assist with decisions aimed at best practice and may also help to conceptualize clinical trials to close relevant knowledge gaps.

It is widely held that parathyroid cancer is an often or eventually fatal condition, and that little can be done to improve outcomes [21,22]. Progress in systemic therapy remains limited [49,63] despite our growing understanding of the underlying mutational landscape [64,65,66,67]. We here provide further data to strengthen the argument that a tangible improvement of outcomes depends on surgical performance, a fact also emerging from rare available multicenter studies [48].

Our study presents a sizable cohort with an unambiguous diagnosis of parathyroid cancer. All cases were reviewed by an expert endocrine pathologist (SDC) and discussed at our multidisciplinary endocrine cancer team meetings. The clinical approach markedly varied within the cohort. A subgroup of patients was referred only after the initial treatment episode had already concluded elsewhere. Here, the potential diagnosis of parathyroid cancer had not been recognized. Accordingly, patients underwent local resection without lymphadenectomy resulting in frequent R1 margins and common loco-regional recurrence. Delayed revision surgery was then delivered at the center. The majority of patients, however, emerged from a large cohort of patients referred for surgical treatment of primary hyperparathyroidism (pHPT). In our dedicated endocrine surgery clinic, we stringently explored these patients for potential parathyroid malignancy at the pre-operative stage. We scrutinized clinical and laboratory features and used expert in-house ultrasonography assessment applying defined criteria [3,68,69]. If parathyroid malignancy was suspected, patients underwent surgery by an experienced endocrine surgeon (KMS) using a pre-defined surgical approach aimed at avoidance of margin failure and eventual regional lymph node clearance [3,23,24]. These patients consistently underwent en bloc resection and margin clearance was achieved in all but 17.6% of these patients. Occasionally, adherence to oncological criteria proceeded at the cost of planned resection of the recurrent laryngeal nerve (RLN), a measure explicitly pre-agreed with the patients during the consent procedure. During the duration of this cohort observation only a single patient with wrongly suspected parathyroid cancer underwent planned RLN resection and was then found to exhibit benign disease (<0.1%).

All patients were scheduled for strict 3-monthly follow-up in our endocrine surgical clinic, obtaining PTH and corrected calcium on all occasions. We followed up on any abnormal results with a full gamut of diagnostic approaches and eventually with radical redo surgery followed by external beam radiation. No patient exhibited lymph nodal recurrence.

The long-term data presented here stand out against published data from larger case series and nation-wide register studies (Table 7). The disease-specific survival rate of 100% at 5 and 10 years is 10–20% higher than in comparable cohorts or nation-wide register studies. There is only a single cancer-related death, an observation different from register studies from the United States, but in line with observations from the largest European multi-center cohort study [59].

In our cohort, overall survival at 10 years is 16% higher in absolute terms (Table 7): Our series shows a 10-year OS of 84.2%, contrasting a 10-year OS of 67.1% in the SEER database [13]. The observation is particularly interesting because the cohorts are demographically similar, as the mean age of both cohorts is 62 years [11], with similar rates of regional metastatic spread (SEER 17.2% [11]) and distant metastasis (SEER 2.2% [11]). In both cohorts, tumour size was under 4 cm in 95% of patients (SEER 94.3% [11]), and surgery was the overwhelmingly used therapy modality.

It is questionable whether comparison between a particular cohort and large cohorts from registers is able to sustain a tenable argument about the causation of eventual outcome differences. We therefore forego this discussion and instead offer the insights emerging from analysis within our cohort. Table 1, Table 2, Table 3 and Table 4 offer an indication of eventual outcome drivers, pointing at initial failure to control disease as sentinel event.

The power of eventual conclusions is necessarily restricted by the small cohort size, yet does the analysis offer a practical approach to understand how the clinical-surgical factors, rather than just the extent or aggressiveness of the underlying tumour, might have shaped outcomes. The body of evidence here presented suggests that size or aggressiveness of the cancer may take second place in the determination of outcome if oncological practices are used to overcome them.

Good survival outcomes appear to be driven by three different lines of approach: (a) scrutiny for potential malignancy and appropriately aggressive surgery to achieve R0 resection (b) close follow-up for early detection and timely radical redo surgery in cases with local recurrence, (c) use of adjuvant radiotherapy when the risk of recurrence is deemed high, either because resection margins are not free following resection of the primary, or because they can no longer be safely established in the setting of revision surgery, a condition with potentially multifocal local recurrence.

### 4.1. A High Index of Suspicion Drives Intra-Operative Decision-Making and Promotes Clear Margins

Our series provides granular detail regarding the intention to cure parathyroid cancer as opposed to suspected benign parathyroid disease, based on pre-operative recognition of malignant potential. An upfront curative surgical approach requires a high index of pre-operative suspicion based on a number of criteria. All patients were clinically assessed for a history compatible with hyperparathyroidism jaw-tumour syndrome or MEN1, eventually including genetic screening [3,70,71]. We considered any lesion potentially malignant when pre-operative ultrasound revealed a size >3.0 cm or when the corrected serum calcium level exceeded 3.0 mM [3,5]. All patients underwent expert ultrasound assessment [68,69], and eventually MIBI-SPECT imaging [72].

Different from large multicenter cohorts [7,59] or nationwide register data [11,13,53,58,60,61], we could apply stringent surgical quality controls in this single-surgeon series. Data from the National Cancer Data Base (NCDB) of the United States, for example, reveal that cancer removal was incomplete in 65.7% of 733 patients [7], with radical surgery performed in only a minority of patients documented on the SEER database [60], and en bloc resection performed in less than 10% of patients in contemporary US data from SEER [53].

Some series identify positive resection margins (R1) in all patients where this criterion has been assessed [54], whilst a major register study from the United States including 1022 patients [10] identified positive margins (R1) in 183/784 or 23.3% of patients with known status. In a subgroup analysis of patients with PC as their only malignancy, positive margins—as well as older age, black race, >2 comorbidities, positive lymph nodes—demonstrated an increased risk of death at 5 years and a lower OS (*p* < 0.05 for all) [10].

Table 3 shows that the use of oncological resection techniques significantly reduces the risk of positive margins by 72% (*p* < 0.05), and the risk of recurrence by 86% (*p* = 0.08). This is in line with multiple studies identifying surgical under-performance by “parathyroidectomy alone” as major risk factor for mortality on univariate and multivariate analysis [52], whilst the effect is elusive in data from the NCBD [10] and SEER registers.

Table 4 identifies that failure to achieve cancer-free margins increases the risk of loco-regional recurrence 18-fold (*p* < 0.05), which corroborates common wisdom hitherto based on sparse supporting data [23].

We argue that a high index of suspicion identifies most patients with parathyroid cancer, that pre-operative recognition promotes adherence to well-established rules of oncological surgery, and thereby enhances the rate of R0-resection to consequently reduce the risk of local recurrence.

One could consider a shift of current recommendations by the American Association of Endocrine Surgeons Guidelines for Definitive Management of Primary Hyperparathyroidism [20], in as far as these guidelines make no recommendations conducive to the pre-operative identification of parathyroid cancer, beyond the general comment on genotype-phenotype relations (recommendation 2-2) and referral to experts capable of deciding about best localization modalities (recommendation 4-2). Notably, the focus of these guidelines appears to be on intra-operative recognition of parathyroid carcinoma, at which point ‘complete resection avoiding capsular disruption is proposed to improve the likelihood of cure as it may require en bloc resection of adherent tissues (strong recommendation; low-quality evidence) (recommendation 12-2)’. We argue that intraoperative recognition of cancer not only requires expert skills, but often comes too late in the course of surgery, namely when dissection focally turns out to be unexpectedly difficult. At that point, the resection margin may have already been lost, and microscopic remnants may cause recurrence. We explicitly concur with said guidelines that recognition of malignancy and oncological performance is key to eventual cure, but seek to shift the vent of recognition to the pre-surgical stage, wherever possible.

This view should replace continued controversy with claims that local resection might be sufficient [36,44,47,50,73,74,75,76,77,78,79,80]. Given the infrequent performance of assuredly oncological surgery in patients in the SEER database, claims that oncological surgery does not significantly improve prognosis (although authors identify differences in cancer-specific survival and overall survival) [80] should be viewed with caution. Rather than trying it on, patients with a suspicion of malignancy should be referred to an expert team, ensuring access to an expert surgeon. There is reason to expect that the volume-outcome associations for parathyroid surgery in England from data of the UK-wide Getting It Right First Time Program pertain not just to benign disease [81], but also benefit malignancy as here shown.

Only thorough pre-operative assessment and recognition of an eventual risk of malignancy can properly inform consent. As surgeon and patient embrace the purpose of curing malignancy, rather than merely normalizing calcium excess, they agree an adapted surgical-oncological strategy. Intraoperative decision-making is then based on the full scope of trade-offs, rather than focal hesitation to avoid “excessive surgery without mandate”, guilt from “unwarranted loss” of the recurrent nerve, or even litigation for deviation from the pre-agreed scope of surgery.

### 4.2. The Limited Role of Second-Line Management

The concept of cancer survivorship increasingly focuses on outcomes beyond mere survival. We here demonstrate a significantly higher treatment burden in patients in whom free resection margins are not achieved during the initial surgery (Table 6). Whilst the association of failed disease control with higher treatment needs is self-evident, it is important to recognize that disease control is a function of the prior discussed first-line management. This puts an oncological approach achieving clear margins as top priority.

Albeit it is clear that positive margins should be interpreted as signum mali ominis, we demonstrate that not all is lost when they occur. Strict 3-monthly follow-up, with attendance eventually reinforced by the dedicated endocrine cancer nurse, led to timely discovery of recurrence and surgical revision in our cohort. We note that all loco-regional recurrences had occurred with three years of surgery (Figure 1; Appendix A). This coincides with observations by other case series [8,48,51], which otherwise do not provide granular data regarding the timing of local versus any recurrence. The observation argues in favor of systematic follow-up as proposed in guidelines [20], and we propose to emphasize that follow-up should be frequent (4×/year) in the first 3–5 years. This also aligns with observations of survivorship across the wider panel of cancers with which parathyroid malignancy clusters: these cancers exhibit a high-risk period of recurrence for some 2–3 years after surgery during which locoregional recurrence outpaces and outnumbers systemic spread [12].

When faced with the need of redo surgery, surgical expertise is of the essence. All our cases undergoing redo surgery remained free of local recurrence during a long follow-up period. That is in line with observations with beneficial outcomes of extended redo operations in the few available larger series [44]. Wei and colleagues showed that 5-year overall survival (OS) following extended en bloc resection during revision surgery was 59.6%, compared with 16.7% after less radical procedures. Their series does not employ adjuvant radiotherapy, indicating that extensive surgery alone may be sufficient to achieve good outcome unless distant metastases are present.

### 4.3. External Beam Radiation Therapy as Adjunct to Surgery?

In a number of cases we employed hypofractionated external beam radiation therapy (EBRT), applying 20–30 fractions to deliver a total dose of 55–60 Gy to a field encompassing the central and lateral neck from jawline to chest aperture and in line with regimens used in endocrine cancers [82]. In absence of a prospective study protocol, case-based decision-making in the multidisciplinary endocrine cancer MDT prioritized this adjuvant modality for patients displaying R1 margins or having undergone revision surgery. The probability of receiving EBRT was 22-fold higher in patients with R1 margins (*p* < 0.05) (Table 6). This significantly added to therapy burden in patients with positive margins.

Defining the finding of either a positive margin after initial surgery or the state following redo surgery as “scenario at risk” we identified 12 such scenarios. In patients with such high risk of loco-regional recurrence Kaplan–Meier survival analysis found patients receiving EBRT to fare significantly better (Figure 2). On the contrary, the benefit of EBRT remained insignificant in risk analysis (Table 5). The apparent discrepancy is explained by differences in the methodological approach: the final estimate in survival analyses is based on calculation of multiple successive probabilities, rather than the probability at a single time point [83]. Given the lasting suppression of local recurrence following EBRT, one might argue that these patients were cured of loco-regional disease. Our study is underpowered to afford firm conclusions, but our data converge with those from MD Anderson Cancer Centre, where administration of a similar EBRT regimen in six of their patients was followed by only a single recurrence [32]. A further report from the same center includes eight patients, with similarly encouraging results and a cautious recommendation to consider EBRT when there is concern for locoregional recurrence [56]. Data from large registers such as the National Cancer Database NCDB [84] do not show a prognostic benefit from EBRT [7], including in a detailed analysis published 2021 [85]. The main criticism of such analyses is the use of “overall survival” as endpoint. Overall survival is dramatically different from locoregional recurrence free survival, i.e. the direct benefit of locoregional EBRT. In how far better locoregional control by EBRT will translate into better disease-specific survival entirely depends on the contribution of untreated local recurrence to the advent of distant metastasis. At the same time, all major datasets prove that overall survival (OS) is not a good surrogate for disease-specific survival (DSS) in parathyroid cancer as most patients die from other causes. Rather, overall survival is dominated by secondary cancers, cardiac events, and other causes (Table 3, risk cluster 1 in [12]). However, our own analysis of further published data regarding adjuvant EBRT fails to convincingly prove the benefits of EBRT (Table 8). As true for all cohort studies, and particularly for those with small sample size, no further conclusions should be drawn before future studies eventually corroborate our observations. We believe that radiation therapy, often used only late in the course of disease, may simply come too late to mitigate such systemic spread. Whilst we do not believe that the data presented here are sufficient to endorse a change in current guidelines, which advise to avoid adjuvant radiation [20], they should encourage the conceptualization of a multi-center randomized-controlled study exploring the impact of adjuvant radiation therapy in a select cohort of patients with positive margins, using loco-regional disease control as primary outcome.

## 5. Conclusions

Acknowledging limited cohort size, our data provide granular detail and significant evidence that pre-operative scrutiny paired with adequately aggressive expert surgery can achieve margin-free resection in most patients with parathyroid cancer, that such margin-free resection translates into improved locoregional disease control, affording long-term cancer-free survival. The suggestion that adjuvant radiation therapy may benefit patients with positive margins deserves dedicated future study.

In summary, we propose to abandon a pessimistic perspective unconducive to expert team referral that leaves work-up and intra-operative course to makeshift decisions. It is time to get it right the first time more often and acknowledge parathyroid cancer as a mostly curable or at least long-term survivable condition, benefiting from enhancement of all aspects of care provision to transform cancer survivorship [86].

## Figures and Tables

**Figure 1 cancers-15-00199-f001:**
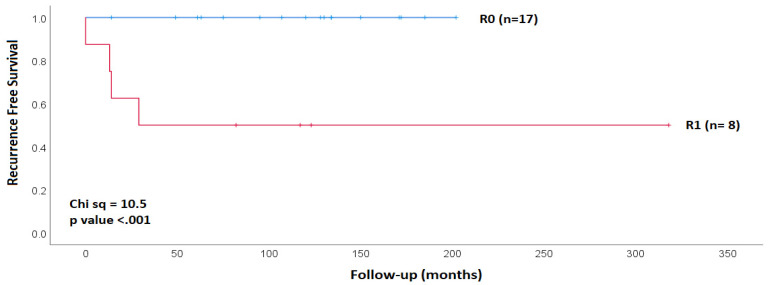
Kaplan–Meier plot of recurrence free survival (RFS) according to margin status following initial surgery. RFS was 100% in patients with R0 resection and 50% in patients with R1 resection.

**Figure 2 cancers-15-00199-f002:**
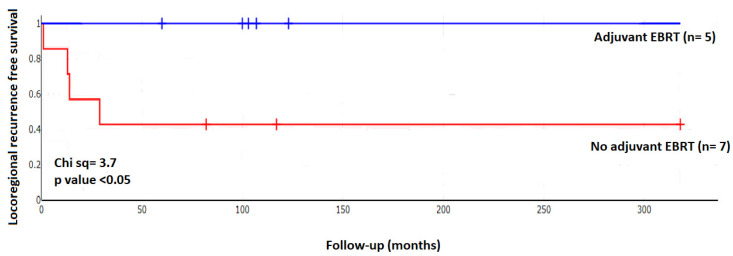
Kaplan–Meier plot of locoregional recurrence free survival following adjuvant EBRT; χ^2^ = 3.72 *p* = 0.05. In 8 patients with R1 resection margins following initial surgery, 12 situations of a risk of recurrence arose either at the time of initial surgery (R1 margin, *n* = 8) or at the time of first revision surgery (*n* = 4). Adjuvant EBRT (55–60 Gy in 20–30 fractions) was administered (Group 1) in 5 out of 12 at risk scenarios, whilst 7 scenarios were followed up without intervention (Group 0). Patients were followed up for 5–26 years following risk exposure.

**Table 1 cancers-15-00199-t001:** Baseline characteristics of patients undergoing R0 and R1 resection.

Clinical Characteristics	R0	R1	Chi Square
	*n* = 17	*n* = 8	*p* Value
gender			
male/female	5/12	5/3	n.s.
age (years)			
mean ± SD	54.4 ± 14.5	62.9 ± 11.9	n.s.
median	58.0	63.5	
range	33–82	43–81	
PTH ^a^			
mean ± SD	9.3 ± 7.2	7.1 ± 4.6	n.s.
median	6.1	6.7	
range	1.5–22.3	1.3–13.7	
corrected calcium (mmol/L)			
mean ± SD	3.1 ± 0.5	3.1 ± 0.3	n.s.
median	2.8	3.0	
range	2.8–4.7	2.7–3.6	
size of lesion (mm)			
mean ± SD	38.4 ± 15.3	31.4 ± 14.1	n.s.
median	35.0	25.0	
range	15–65	20–56	
lymph node metastasis			
yes/no	0/14 ^b^	1/3 ^c^	n.s.
distant metastasis			
yes/no	0/17	0/8	n.s.
histology			
low risk/high risk	9/8	5/3	n.s.

^a^: Calculated as times the upper normal limit of the method utilised. ^b^: lymph node dissection was not done in 3 patients (1 patient had initial local excision elsewhere). ^c^: lymph node dissection was not done in 4 patients (3 patients had initial local excision elsewhere). n.s: not statistically significant.

**Table 2 cancers-15-00199-t002:** Regression analysis for prediction of recurrence free survival (RFS).

Clinical Characteristics			Univariate Analysis	Multivariate Analysis
	RFS	No RFS	Chi Square	Chi Square
	*n* = 21	*n* = 4	*p* Value	*p* Value
gender				
male/female	5/16	3/1	n.s.	n.s.
age (years)				
mean ± SD	59.9 ± 14.0	67.0 ± 10.0	n.s.	n.s.
median	59.0	65.0		
range	33–82	57–81		
PTH ^a^				
mean ± SD	9.0 ± 6.8	6.7 ± 4.5	n.s.	n.s.
median	6.7	6.6		
range	1.5–22.3	1.3–22.3		
corrected calcium (mmol/L)				
mean ± SD	3.1 ± 0.5	3.1 ± 0.4	n.s.	n.s.
median	2.9	3.1		
range	2.8–4.7	2.7–3.6		
size of lesion (mm)				
mean ± SD	36.9 ± 12.7	32.8 ± 12.7	n.s.	n.s.
median	33.0	30.5		
range	15–65	20–50		
margin status	17/4	0/4	10.1	19.5
R0/R1			<0.01	<0.001

^a^: Calculated as times the upper normal limit of the method utilised.

**Table 3 cancers-15-00199-t003:** Initial surgery: resection choice predicts margin status and recurrence.

Surgery	All	Margin			Recurrence		
		R1 Positive	R0 Negative	Positive Margin	Yes	No	Risk of Recurrence
all	25	8	17	32.0%	4	21	8.0%
en bloc resection	17	3	14	17.7%	1	16	5.9%
local excision	8	5	3	62.5%	3	5	37.5%
relative risk (RR)				3.54			6.38
95% CI				(1.11–11.28)			(0.78–52.1)
significance				*p* = 0.03			*p* = 0.08

**Table 4 cancers-15-00199-t004:** Initial margin status informs loco-regional recurrence risk.

Margin	All	Recurrence	Recurrence	Risk of Recurrence
		Yes	No	
positive	8	4	4	50.0%
negative	17	0	17	0%
relative risk				18.0
95% CI				1.1–299.0
significance, two-sided				*p* = 0.04

**Table 5 cancers-15-00199-t005:** Adjuvant EBRT reduces local recurrence risk in patients at high risk of locoregional recurrence (defined as R1 margin status after initial surgery or status post redo surgery).

		Recurrence	Recurrence	Risk of Recurrence with EBRT
		Yes	No	
all	12	4	8	33.3%
adjuvant EBRT	5	0	5	0.0%
no EBRT	7	4	3	57.1%
relative risk				0.18
95% CI				0.01–2.65
significance, two-sided				*p* = 0.20
Odd’s ratio				0.09
95% CI				0.003–2.203
significance, two-sided				*p* = 0.13
Number needed to treat NNT to achieve a benefit				2.2
95% CI				1.1–51.4

**Table 6 cancers-15-00199-t006:** Survivorship: therapy burden beyond the initial surgery depends on margin status after initial surgery.

Parameter	R0 Resection	R1 Resection	Difference
	*n* = 17	*n* = 8	
redo surgery	0	4	RR 18.0
			95% CI 1.1–299.0
			*p* = 0.04
adjuvant EBRT	0	5	RR 22.0
			95% CI 1.4–355.5
			*p* = 0.03
recurrent nerve palsy (following nerve resection)	3	1	RR 1.41
			95% CI 0.2–11.6
			*p* = 0.75
days of all inpatient admissions (mean ± SD) ^a^	0	8 (1.0 ± 1.1)	not calculated
days of outpatient admissions (mean ± SD) ^b^	4 (0.3 ± 1.0)	176 (22.0 ± 23.7)	Chi sq = 979.5 *p* < 0.001
“days of worry”; i.e., days before either - PTH and calcium normalized by surgery ^c^ - adjuvant radiotherapy completed ^d^	0	1369 (171.1 ± 302.5)	Chi sq = 11.3 *p* = 0.003

^a^ Count of days of disease-related inpatient admission AFTER completion of the initial surgical episode: eventual complications, redo surgery, hypercalcaemia requiring in-hospital treatment. ^b^ Count of days of outpatient admission; e.g., each fraction of radiotherapy is *n* = 1 outpatient admission. ^c^ Count of days between initial surgery and definitive normalisation of PTH and corr. calcium. For the patient with regional persistent disease (No 18) this includes nearly 700 days (2010–2012), but not days after diagnosis of metastatic disease. For patients with recurrent disease, we counted the number of days between diagnosis of recurrence until redo surgery. ^d^ Where adjuvant radiation therapy was provided, we counted days after initial or redo surgery until completion of radiation therapy.

**Table 7 cancers-15-00199-t007:** Survival across Parathyroid Cancer Cohorts.

Source	*n*	Overall Survival		Recurrence Free Survival		Disease-Specific Survival	
		OS		RFS		DSS	
		5-Year	10-Year	5-Year	10-Year	5-Year	10-Year
case series							
range	484	50.0–100%	43.0–100%	33.2–88.9%	29.4–83.0%	80.0–100%	72.0–100%
median		79.2%	68.0%	62.0%	51.0%	90.9%	83.3%
register data							
range	1703	78.5–90.6%	49.1–72.9%	59.6–79.0%	51.5–70.8%	82.5–94.1%	67.0–92.1%
median		82.6%	68.3%	73.9%	69.1%	87.0%	78.9%
our cohort	25	95.8%	84.2%	82.6%	79.0%	100%	100%
R0 or R1 + EBRT	19	94.7%	89.4%	100%	100%	100%	100%
R1, no EBRT	6	100%	66.7%	33.3%	33.3%	100%	100%

Case series contain data from 18 studies [9,28,29,30,31,32,33,34,35,36,37,38,39,40,41,42,43,44]; detail is provided in Appendix A. Register data contain data from 16 reports [7,8,10,11,13,35,39,45,46,47,48,49,50,51,52,53], considering overlapping data from re-reports of the same register only once; detail is provided in Appendix A. Full data for our own cohort are provided in Appendix A. The subcohort “R0 or R1 + EBRT” comprises all patients which either had a negative margin status after initial surgery or had positive margins (R1) and received adjuvant EBRT after initial surgery. The subcohort “R1, no EBRT” comprises of patients with positive margin status (R1) after initial surgery and who received no EBRT at that stage.

**Table 8 cancers-15-00199-t008:** Outcomes following adjuvant radiotherapy after initial surgery reported in the literature.

Year	Author	*n*	EBRT	No EBRT
			Recurrence	Recurrence
			Yes/No	Yes/No
1998	Chow [29]	10	0/6	3/1
2003	Munson [31]	4	0/4	0
2004	Busaidy [32]	26	1/5	9/11
2011	Schaapveld [46]	10	3/1	5/1
2013	Erovic [54]	16	3/2	2/9
2013	Selvan [55]	9	0/6	3/0
2017	Christakis [56]	8	1/4	3/0
2019	Akirov [57]	7	4/0	2/1
2021	Sali [42]	16	2/7	0/7
2021	Cunha [43]	17	3/0	5/9
1999–2014	Total	123	17/35	32/39
			32.7%	45.1%
relative risk of recurrence with EBRT				0.7
95% CI				0.45–1.16
significance				*p* = 0.18

## Data Availability

The authors confirm that the findings of this study are available within the article and its Appendix A. Raw data that support the findings of this study are available from the corresponding author, upon request.

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
