# Peer review of "Margin Free Resection Achieves Excellent Long Term Outcomes in Parathyroid Cancer"

_cancers, 2022, doi:10.3390/cancers15010199_

Round 1

Reviewer 1 Report

The authors have attempted to shed light on the issue of an oncologic surgical approach to parathyroid cancer, which already remains poorly documented.

The case series presented here provides evidence that an appropriate oncologic surgical approach (R0, en bloc resection with removal of the central compartment of lymph nodes) can bring lasting benefits to patients, giving a 10-year disease-specific survival rate of 100%.  The group has published a smaller series in the past, and this article includes more patients and provides new follow-up and survival data. It is interesting to note that the use of EBRT in patients with R1 resection did not result in parathyroid cancer recurrence.

Although the numbers in this study are small, there is not much data in the literature on this group of patients and any new data is interesting.

Certainly a larger study is required and it would indeed be interesting to see if this benefit continues to be observed.

Overall, I would recommend this article for publication because of the clinical relevance and the interest of the endocrinology world in this clinical situation.

Author Response

Dear Reviewer 1

thank you indeed for your review and support of this manuscript.

Merry Christmas and a Happy New Year 2023

Klaus-Martin Schulte 

Reviewer 2 Report

I am really admired with your well-documentated study, because it offers optimistic approach to parathyroid carcinoma, being often or eventually fatal condition.

Author Response

Dear Reviewer 2

thank you indeed for your review and support of this manuscript.

Merry Christmas and a Happy New Year 2023

Klaus-Martin Schulte 

Reviewer 3 Report

A very interesting paper with a long follow-up. 
It is not possible to se the Supplementary tablets in order to see in detail how your cohort distinguishes from previously published case reports. 
Could you add a Baseline Table in the manuscript? 

Author Response

Dear Reviewer 3,

thank you indeed for your careful review and interest.

We have addressed your items. See word file. 

Merry Christmas and a Happy New Year 2023

Reviewer 4 Report

The authors evaluated long-term outcomes of patients with parathyroid cancer at a medical center. They found En bloc resection achieved cancer-free resection margins (R0) in 82.4% and loco-regional disease control in 94.1% patients. On the contrary, local excision exhibited poorer outcomes with a significant 3.5-fold higher risk of positive margins than en bloc resection (p=0.03), and having statistically insignificant 6.4-fold higher risk of locoregional recurrence (p=0.08). There are some comments on this draft.

1.     Complete resection of parathyroid cancer (R0) is associated with better overall survival in a report as the authors mentioned (lines 414-416). Therefore, the data presented in this draft is not novel.

2.     In this study, 17 patients received En bloc resection and 8 patients received local excision for the treatment of parathyroid cancer. The authors may clarify the reasons for the choice of surgical approaches in each patient. Confounding by indication may lead to different outcomes.

3.     The basic demographic data of these 25 patients may be summarized in a Table to address the clinical features of these patients.

4.     To demonstrate the effects of treatment, patient characteristics, including age, sex, calcium level, tumor size, lymph node involvement and distant metastasis should be included for univariant and multivariant analysis. These analyses are not presented in this draft.

5.     EBRT was not associated with lower risk of recurrence of parathyroid cancer (P=0.20) in Table 4. However, EBRT was significantly associated with lower risk of recurrence of parathyroid cancer (P<0.05) in Figure 2. It is not clear if EBRT was correlated with better outcomes according to these data. The reasons accounting for this discrepancy may be clarified.

6.     The Kaplan-Meier curves of overall survival and disease-specific survival in 25 patients should be analyzed to have clear information regarding the outcomes of these patients.

7.     One patient had locally advance disease, did this patient achieve R1 or R2 resection?

8.     This draft was not well written and should be improved.

Author Response

Dear Reviewer 4,

thank you very much indeed for your careful and detailed review. We have addressed all issues point by point. Please see attached.

A Merry Christmas and a Happy New Year 2023

Round 2

Reviewer 4 Report

The authors have adequately addressed the points I raised. I have no further comments.